# Return of the Latent Space COWBOYS: Re-thinking the use of VAEs for Bayesian Optimisation of Structured Spaces

Henry B. Moss [1 2]    Sebastian W. Ober [3]    Tom Diethe [4]

## Abstract

Bayesian optimisation in the latent space of a Variational AutoEncoder (VAE) is a powerful framework for optimisation tasks over complex structured domains, such as the space of scientifically interesting molecules. However, existing approaches tightly couple the surrogate and generative models, which can lead to suboptimal performance when the latent space is not tailored to specific tasks, which in turn has led to the proposal of increasingly sophisticated algorithms. In this work, we explore a new direction, instead proposing a decoupled approach that trains a generative model and a Gaussian Process (GP) surrogate separately, then combines them via a simple yet principled Bayesian update rule. This separation allows each component to focus on its strengths— structure generation from the VAE and predictive modelling by the GP. We show that our decoupled approach improves our ability to identify high-potential candidates in molecular optimisation problems under constrained evaluation budgets.

## 1. Introduction

First introduced by Gómez-Bombarelli et al. (2018), Bayesian Optimisation (BO) over latent spaces has emerged as a powerful technique for optimising over structures. Rather than performing challenging combinatorial or high-dimensional optimisation directly on discrete structures — such as molecules or proteins — Latent Space BO (LSBO) first maps inputs into a fixed-dimensional Euclidean latent space, where standard surrogate models and gradient-based

[1] School of Mathematical Sciences, Lancaster University, UK [2] Department of Applied Maths and Theoretical Physics, University of Cambridge, UK [3] Oncology R&D, AstraZeneca, Gaithersburg, USA [4] Biopharma R&D, AstraZeneca, Cambridge, UK . Correspondence to: Henry B. Moss <hm493@cam.ac.uk>.

*Proceedings of the 42nd International Conference on Machine Learning*, Vancouver, Canada. PMLR 267, 2025. Copyright 2025 by the author(s).

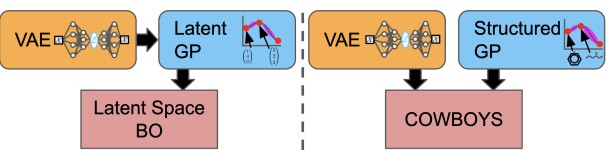

*Figure 1.* Unlike LSBO where GPs are fit in a VAE's latent space, COWBOYS's GP is fit in structure space, decoupled from its VAE. acquisition routines can be employed. Candidate points selected by the optimiser are then decoded back into the original structured domain to yield new query points.

Learning a complex mapping from variable-size structured inputs to fixed-size Euclidean representations requires more data than is typically gathered in LSBO itself. Therefore, LSBO relies on embeddings pre-trained on a related task with abundant unlabelled data (e.g., using a VAE latent space trained on a large set of valid molecules). However, it is well known (Chu et al., 2024) that fitting surrogates in the latent space of a VAE can lead to suboptimal modelling; therefore, the recent LSBO literature has focused on introducing increasingly sophisticated heuristics to fine-tune the VAE on newly acquired experimental data. These approaches remain challenged by the persistent risk of overfitting when training neural network components of Bayesian models on limited datasets (Ober et al., 2021). In this work, we also make a new argument that the prevalent approach of limiting the search to a fixed box subset of the latent space — necessary to apply traditional BO methods — also limits the effectiveness of LSBO.

While the VAE is designed purely as a generative model, LSBO attempts to repurpose it for discrimination — a fundamental mismatch that gives rise to the pathologies discussed above. We propose an alternative approach that preserves the original motivation of the VAE as a generative model by coupling it with a separately trained surrogate GP model via a novel Bayesian formulation. Our primary goal is straightforward: *develop a sampling strategy that increasingly favours structures with desirable objective values.*

We propose Categorical Optimisation With Belief Of underlYing Structure (COWBOYS), a novel principled framework that unifies GPs and VAEs within a BO loop. Although we focus on molecular search—the most popular application of LSBO—where we demonstrate that COWBOYS

excels under heavily constrained optimisation budgets, our approach is general and applicable to any structured problem for which a suitable structural GP kernel can be defined (see our discussion in Section 8).

## 2. Background

### 2.1. Bayesian Optimisation

Bayesian Optimisation (BO) (Mockus, 2005) is a powerful framework for the efficient optimisation of costly black-box functions $f : \mathcal{X} \to \mathbb{R}$. For a successful application of BO, we typically need three components: a search space, a surrogate model, and an acquisition function. Given some initial data, BO constructs a surrogate model of the data, which is then used in tandem with an acquisition function to determine which point in the search space will be most valuable to acquire. Once this point is chosen, we query the black-box function at this point and update the surrogate model, repeating until the evaluation budget is exhausted.

**Search spaces**. In order for optimisation to succeed, the space over which we attempt to find the optimal value must be suitably restricted. For instance, when dealing with a Euclidean space $\mathbb{R}^d$, it is typical to restrict the search space to a box region $[-\delta, \delta]^d$, where $\delta$ is typically chosen using some prior knowledge about the expected location of the optimum (Hvarfner et al., 2022). In this example, choosing too large a $\delta$ will lead to slow convergence due to the need for more data so that the surrogate can be effective, whereas too small a $\delta$ will risk missing the optimum. Later, we will discuss how choosing appropriate search spaces is a key bottleneck for LSBO and how COWBOYS addresses this.

**Surrogate models**. BO requires surrogate models that accurately quantify uncertainty while retaining flexibility: as such, Gaussian Processes (GPs) (Rasmussen, 2003) are a standard choice. A GP can be defined as an infinite collection of random variables, any finite number of which are Gaussian distributed, and is entirely defined by a mean function $\mu(\cdot)$ and a kernel $k(\cdot, \cdot)$. Under a GP $f \sim \mathcal{GP}(\mu, k)$, making a prediction $\hat{f}$ at a test point $\hat{\boldsymbol{x}}$ amounts to conditioning on the observed data $D : p(\hat{f}|D) = \mathcal{N}(\mu^*, \Sigma^*)$, where $\hat{\mu}$ and $\hat{\Sigma}$ can be computed in closed form via the properties of Gaussians with $O(N^3)$ computational complexity and $O(N^2)$ memory.

**Acquisition functions**. Acquisition functions measure the utility of acquiring an unseen point $\boldsymbol{x}$ according to the predictions of our surrogate model. Common acquisition functions measure the Expected Improvement (EI), Probability of Improvement (PI) (Bergstra et al., 2011) or Upper Confidence Bounds (UCB) (Auer, 2000) of candidate points, or use information theoretical arguments (Hennig & Schuler,

2012; Moss et al., 2021), e.g.,

$$\alpha_{PI}(\boldsymbol{x}; D) = \mathbb{P}(f(\boldsymbol{x}) > f^*|D), \qquad (1)$$

where $f^*$ is the best objective value observed so far.

**Covariance (kernel) functions**. In this work, where we must build a surrogate model over the space of molecules, we use the Tanimoto kernel (Tripp et al., 2023):

$$K_T(\boldsymbol{m}, \boldsymbol{m}') = \sigma^2 \frac{\boldsymbol{m} \cdot \boldsymbol{m}'}{\|\boldsymbol{m}\|^2 + \|\boldsymbol{m}'\|^2 - \boldsymbol{m} \cdot \boldsymbol{m}'},$$

where $\boldsymbol{m}$ and $\boldsymbol{m}'$ are molecular *fingerprints* representation vectors of molecules and $\sigma \in \mathbb{R}^+$ is a tunable scaling factor. We follow the advice of Tripp & Hernández-Lobato (2024) and use count-based vectors that count specific molecular features (Landrum, 2013). Structural kernels such as the Tanimoto kernel, but also string kernels (Moss et al., 2020a) and graph kernels (Kriege et al., 2020), can often outperform deep learning alternatives, particularly in low- to medium-data regimes (Moss & Griffiths, 2020; Griffiths et al., 2022).

### 2.2. Variational AutoEncoders

Variational AutoEncoders (VAEs) (Kingma & Welling, 2014; Rezende et al., 2014) are deep generative models that model data through a two-step process: sampling a latent variable $\boldsymbol{z}$ (with dimension smaller than $\mathcal{X}$) from a prior $p(\boldsymbol{z})$, followed by sampling $\boldsymbol{x}$ through a distribution $p_\theta(\boldsymbol{x}|\boldsymbol{z})$, resulting in a generative model $\boldsymbol{x} \sim p_\theta(\boldsymbol{x})$. The latter of these steps is performed by a neural network, known as the *decoder*, with parameters $\theta$. The decoder outputs the parameters of the conditional distribution $p_\theta(\boldsymbol{x}|\boldsymbol{z})$, for instance the mean and variance of a normal distribution, or class probabilities for a categorical distribution. In order to train a VAE, we require the posterior $p_\theta(\boldsymbol{z}|\boldsymbol{x})$. However, as this is intractable, we use a neural network *encoder* with parameters $\phi$ to approximate it through amortised inference: $q_\phi(\boldsymbol{z}|\boldsymbol{x}) = \mathcal{N}(\boldsymbol{\mu}(\boldsymbol{x}), \boldsymbol{\sigma^2}(\boldsymbol{z})I)$, where $\phi$ parameterises $\boldsymbol{\mu}(\cdot)$ and $\boldsymbol{\sigma^2}(\cdot)$. The encoder and decoder are trained jointly through a minibatch-friendly lower bound to the marginal likelihood of the data.

## 3. The Pitfalls of Latent Space BO

In what follows, we assume that we have access to a pre-trained VAE with a $d$-dimensional latent space $\mathcal{Z} = \mathbb{R}^d$, resulting in a decoding distribution $p_\theta(\mathbf{x}|\mathbf{z})$, and imply a data distribution $p_\theta(\boldsymbol{x})$.

### 3.1. Latent Space BO

Latent Space BO (LSBO) (Gómez-Bombarelli et al., 2018) can be viewed as standard BO conducted over an alternative search space — the latent space $\mathcal{Z}$ of a VAE — and using

---

**Algorithm 1** Latent Space Bayesian Optimisation

---

**Input:** budget $N$, init size $N_{\text{init}}$, search bounds $\delta$
Clip search space $\mathcal{Z}_\delta \leftarrow [-\delta, \delta]^d \subset \mathcal{Z}$
**for** $n \in \{1, .., N\}$ **do**
    **if** $n < N_{\text{init}}$ **then**         $\triangleleft$ initial design
        $z_n \sim \text{SpaceFillingDesign}(\mathcal{Z}_\delta)$
    **else**         $\triangleleft$ sequential optimisation
        $z_n \leftarrow \text{argmax}_{z \in \mathcal{Z}_\delta} \; \alpha(z; D^{\mathcal{Z}}_{n-1})$   $\triangleleft$ e.g. Eq. (1)
    **end if**
    Decode chosen latent $x_n \sim p_\theta(x|z_n)$
    Evaluate new molecular structure $y_n \leftarrow f(x_n)$
    Update dataset $D^{\mathcal{Z}}_n \leftarrow D^{\mathcal{Z}}_{n-1} \bigcup \{(z_n, y_n)\}$
    Fit latent space GP on $D^{\mathcal{Z}}_n$
**end for**
**return** Believed optimum across $\{z_1, .., z_n\}$

---

a different surrogate model, one modelling the mapping $g : \mathcal{Z} \to \mathbb{R}$ from latent codes to objective function values with a surrogate $\tilde{g}$. At the $n^{\text{th}}$ optimisation step, LSBO proceeds analogously to standard BO. For example, under the PI acquisition function, LSBO selects a new latent code $z_n$ with highest utility

$$z_n \leftarrow \underset{z \in \mathcal{Z}}{\text{argmax}} \; \mathbb{P}(\tilde{g}(z) > f^* | D^{\mathcal{Z}}_{n-1}),$$

where the surrogate $\tilde{g}$ is trained on the dataset $D^{\mathcal{Z}}_{n-1} = \{(z_i, y_i)\}_{i=1}^{n-1}$ of latent code-evaluation pairs. The result $z_n$ is then decoded and the resulting structure is evaluated on the objective function (see Algorithm 1).

We dedicate the remainder of this section to precisely state the two primary pathological issues caused by this current way of coupling VAEs and GPs in LSBO: i) fitting GPs in VAE latent spaces can lead to poor surrogate models, and ii) it is challenging to define a search space within the latent space so that it reliably targets the most performant regions of the decoder and so leads to the selection and generation of high-quality and useful molecular structures (i.e., avoiding chemically invalid or biologically irrelevant molecules).

### 3.2. LSBO Surrogate Models can be Poor Predictors

The premise behind the efficiency of BO is the ability to build an effective surrogate model by exploiting the smoothness of our objective $f$ over the search space $\mathcal{X}$. However, as demonstrated by many (e.g., Griffiths & Hernández-Lobato, 2020; Moss et al., 2020a; Tripp et al., 2020), the mapping $g$ from the latent space to objective function values can be significantly more challenging to learn with a GP than $f$. Consequently, LSBO often ends up with a poor surrogate model that under-represents variations, over-emphasises sub-optimal areas of the space, and fails to extrapolate the target property to not-yet-evaluated structures — a notion typically referred to by the catch-all term "poor latent space

alignment," generally attributed to two VAE properties:

**1) The VAE is trained offline.** Lack of alignment in LSBO is typically attributed to the purely unsupervised training of VAEs, which focuses solely on reconstructing the (unlabelled) training data of the VAE. In particular, as the highly expressive neural network underlying the VAE can distort local neighbourhoods, any smoothness assumptions that motivate Bayesian optimisation in the original space $\mathcal{X}$—for instance, the continuity of a molecule's performance under small structural perturbations—are unlikely to transfer to smoothness in the latent space of the VAE. The vast majority of recent LSBO work (see Section 6) attempts to fine-tune the VAE during optimisation to improve alignment for the current optimisation objective. However, such approaches are plagued by the fundamental challenge of reliably fine-tuning neural networks on small datasets without over-fitting behaviour. Indeed, as the VAE has a large number of hyperparameters from a Bayesian point of view (i.e., the parameters of the decoder), the VAE's decoder is prone to doing all the heavy lifting, which can lead to overfitting analogous to that described for deep kernel learning in Ober et al. (2021).

**2) The VAE has a stochastic decoder.** The second challenge for GP models in LSBO, one largely ignored until the recent works of Chu et al. (2024); Lee et al. (2025), is the stochasticity of the decoder. Since the decoder is stochastic, a single latent code $z$ can decode to multiple different molecular structures $x$, and thereby a range of objective function values (see Figure 2(a)). A standard GP can only explain such discrepancies as additional observation noise or by over-fitting, limiting the utility of predictions across the rest of the space — a well-known problem that has motivated substantial work on providing GP models that support noisy inputs (e.g., McHutchon & Rasmussen, 2011; Qing et al., 2023a).

*In COWBOYS, we avoid alignment issues entirely by departing from using a GP in the latent space, instead only fitting surrogate models directly in the data-space.*

### 3.3. LSBO Search Spaces are not Uniform Boxes

The latent spaces of VAEs are unbounded, therefore LSBO typically restricts the search to a pre-specified bounded region $[-\delta, \delta]^d \subset \mathcal{Z}$ (see Algorithm 1). This simplification serves two main purposes. First, limiting the search space facilitates the use of established acquisition function optimisation methods. Second, by focusing on a central region hypothesized to be more well-behaved, one can limit the presence so-called "dead regions" in the latent space: areas known to yield unrealistic decodings (Griffiths & Hernández-Lobato, 2020). However, for the VAEs typically considered in molecular design problems, we now show that this intuition is entirely incorrect.

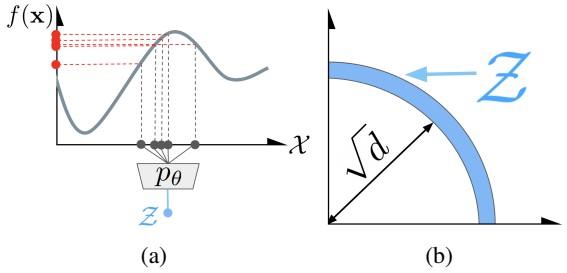

*Figure 2.* (a) In LSBO, the same latent input (blue dot) will, via the stochastic decoder (grey box), map to different values in structure space (black dots) and so corresponds to multiple objective function values (red dots) — a discrepancy that hinders the learning of accurate surrogate models. (b) In higher-dimensional problems, the area of the latent space supported by the prior of the VAE (blue) concentrates in a thin circular shell.

"Dead areas" are regions in a generative model's latent space that lie far from the training data's encodings—encodings which themselves populate high-probability zones under the prior. This concept is often referenced when manipulating latent vectors across a wide range of generative models (White, 2016) (for VAEs and Generative Adversarial Networks (GANs)) and (Song et al., 2020a;b) (for diffusion-based approaches). Outside of the LSBO setting, common strategies for latent space manipulation draw on a classical result which shows that most of a high-dimensional Gaussian's probability mass resides in a thin annulus rather than near the centre:

**Theorem 3.1** (Gaussian Annulus Theorem, Section 3.3.3 of Vershynin (2018)). *Nearly all of the probability mass of a standardised Gaussian is concentrated in a thin annulus of width $O(1)$ at radius $\sqrt{d}$.*

Although the Gaussian Annulus Theorem is well known in machine learning—leading to "radius-preserving" methods for latent value manipulation like spherical linear interpolation (SLERP) (Shoemake, 1985) and sub-space extraction (Bodin et al., 2024)—its implications pose a serious challenge to the practice of clipping the latent search space in LSBO when considering VAEs with even just moderate dimensionality. For example, the encoder of the 128-dimensional VAE used for our second set of experiments (Section 7.2) maps 95% a 5,000 random subset of its training data to be within a 128-dimensional spherical shell of thickness 0.06. Even advanced LSBO approaches, such as adaptive centring of the search space (Maus et al., 2022a), still rely on box-shaped regions, making it impossible to target only this high-probability "shell" where the VAE encoder produces its most useful structures.

*In COWBOYS we can avoid the need to define a search space entirely, instead proposing a sampling-based strategy rather than one requiring acquisition optimisation.*

## 4. Return of the Latent Space COWBOYS

As in LSBO, COWBOYS relies on two probabilistic models: the VAE $\boldsymbol{x} \sim p_\theta(\boldsymbol{x})$ for generating likely valid molecules, and a GP-based predictive distribution $p(f(\mathbf{x})|D_n^{\mathcal{X}})$ for estimating objective values. However, rather than predicting these values directly from latent codes—an often challenging task (see above) — we leverage GP's proven effectiveness in the original structured space using a dataset $D_{n-1}^{\mathcal{X}} = \{(\boldsymbol{x}_i, y_i)\}_{i=1}^{n-1}$ of structure-evaluation pairs and a Tanimoto kernel (as introduced in Section 2.1).

We now summarise Categorical Optimisation With Belief Of underlYing Structures (COWBOYSs), as Algorithm 2. Notably it avoids any explicit reference to the latent space $\mathcal{Z}$. Despite its simplicity, COWBOYS's formulation allows us to exploit the power of the VAE whilst avoiding LSBO's two core challenges: (i) we do not fit models in the latent space and (ii) we do not use the latent space as a search space. To "focus" the generating distribution $p_\theta(\boldsymbol{x})$ towards promising regions during optimisation, we apply a Bayesian update informed by the GP surrogate. The algorithm proceeds via two main steps, described below for the $n^{th}$ iteration under an initial design of size $N_{\text{init}}$.

1. **Initial design** ($n \leq N_{\text{init}}$) To generate our initial design of size $N_{\text{init}}$ we use the VAE exactly as it was designed, i.e. by decoding Gaussian latent samples. Therefore for $n \in \{1, ..., N_{\text{init}}\}$ we sample

$$\boldsymbol{x}_n \sim p_\theta(\boldsymbol{x}), \qquad (2)$$

   resulting in an initial dataset of structure-evaluation pairs over which we can initialise our GP model, without the need to specify a clipped search space.

2. **Optimisation steps** ($n > N_{\text{init}}$) After the initial design, COWBOYS refines its search by only sampling molecules that the surrogate GP predicts will exceed the best observed objective value so far. More concretely,

$$\boldsymbol{x}_n \sim p_\theta(\boldsymbol{x}|f_{\boldsymbol{x}} > f^*, D_n^{\mathcal{X}}), \qquad (3)$$

   where $f_{\boldsymbol{x}}|D_n^{\mathcal{X}}$ is a random variable following the GP's posterior predictive distribution for the objective value at $\mathbf{x}$, and $f^*$ denotes the highest observed objective value so far.

To build intuition for COWBOYS, one might consider a simple, though computationally prohibitive, approach via rejection sampling: generate a large number of candidate structures from the VAE and retain only those satisfying $f_{\boldsymbol{x}} > f^*|D_n^{\mathcal{X}}$ (under a realisation of $f_{\boldsymbol{x}}|D_n^{\mathcal{X}}$).

**Batch BO**. Note that an arbitrary number of samples can be drawn in parallel from (3), making COWBOYS well-suited for the large-batch evaluations often tackled in Bayesian

---
**Algorithm 2** COWBOYS
---

   **Input:** Budget $N$, init size $N_{\text{init}}$
   **for** $n \in \{1, .., N\}$ **do**
      **if** $n < N_{\text{init}}$ **then**           $\triangleleft$ initial design
         $\boldsymbol{x}_n \sim p_\theta(\boldsymbol{x})$           $\triangleleft$ vanilla VAE
      **else**           $\triangleleft$ sequential optimisation
         $f^* \leftarrow \max_{i=1,..,n-1} y_i$
         $\boldsymbol{x}_n \sim p_\theta(\boldsymbol{x}|f_{\boldsymbol{x}} > f^*, D_{n-1}^{\mathcal{X}})$$\triangleleft$ GP-conditioned VAE
      **end if**
      Evaluate new structure $y_n \leftarrow f(\boldsymbol{x}_n)$
      Update dataset $D_n^{\mathcal{X}} \leftarrow D_{n-1}^{\mathcal{X}} \bigcup \{(\boldsymbol{x}_n, y_n)\}$
      Fit structured space GP on $D_n^{\mathcal{X}}$
   **end for**
   **return** Believed optimum across $\{\boldsymbol{x}_1, \ldots, \boldsymbol{x}_n\}$

---

optimisation (Vakili et al., 2021) and active learning (Ober et al., 2025).

# 5. Practical Sampling for COWBOYS

Although the GP-conditioned VAE is a conceptually appealing way to focus on increasingly narrower, relevant proportions of molecular space, sampling from (3) remains non-trivial. Indeed, the naive rejection sampler that uses the VAE $p_\theta(\mathbf{x})$ as a proposal becomes prohibitively inefficient once the region of the search space likely to satisfy $f_{\mathbf{x}} > f_n^*$ narrows, which typically occurs in the early stages of the optimisation process. Consequently, we recommend leveraging more advanced sampling algorithms from the computational statistics literature.

In the remainder of this section, we illustrate how a popular Markov Chain Monte Carlo (Brooks et al., 2011, MCMC) approach can be used to approximate COWBOYS. Specifically, we demonstrate that sampling from COWBOYS' GP-conditioned Variational AutoEncoder (VAE) can be reformulated as sampling from a posterior distribution induced by a Gaussian prior and a corresponding likelihood — a well-established setting for which effective MCMC algorithms already exist and can be readily applied.

## 5.1. COWBOYS $\approx$ Inference under a Gaussian Prior

Our practical implementation of COWBOYS approximates the GP-conditioned VAE (3) as the posterior distributions induced by a specific likelihood and a (potentially high-dimensional) Gaussian prior. However, this approximation strategy requires a deterministic mapping from the latent space to the structure space, rather than the stochastic mapping provided by the VAE's decoder. Consequently, COWBOYS only ever considers the most likely decoding of the latent variable, inducing a deterministic mapping between latent codes and structure given by $h_\theta(\boldsymbol{z}) =$

$\text{argmax}_{\boldsymbol{x}}\, p_\theta(\boldsymbol{x}|\boldsymbol{z})$. In other words, we approximate the VAE's decoder distribution with the delta function:

$$\hat{p}_\theta(\boldsymbol{x}|\boldsymbol{z}) \approx \delta(\boldsymbol{x} - h_\theta(\boldsymbol{z})),$$

i.e. a deterministic decoding strategy that selects only the most-likely decoded molecule from each latent location $\mathbf{z}$.

Note that we are not the first to replace the VAE's probabilistic decoding with the most-likely mapping (e.g., González-Duque et al. (2024)). Removing this source of variation can, at the risk of reducing the diversity of candidate samples, mitigate the alignment issues discussed in Section 3.2. In our experiments, however, we found that COWBOYS did not suffer empirically from this reduced variation (see Appendix B). We also stress that the VAE is still initially trained with a stochastic decoder, and that it is only when performing BO that we take the most likely mapping.

Once we have defined the mapping $h_\theta : \mathcal{Z} \to \mathcal{X}$, each latent variable $\boldsymbol{z} \in \mathcal{Z}$ now defines a corresponding structure $\boldsymbol{x} \in \mathcal{X}$. Consequently, rather than sampling a new structure directly, we can reframe the sampling task in (3) as sampling a new latent value $\boldsymbol{z}$ from an appropriate distribution and then mapping $\boldsymbol{z}$ through $h_\theta$ to obtain the new structure. Specifically, under the deterministic approximation of the decoder distribution, sampling $\boldsymbol{x}$ from (3) is equivalent to the following two step procedure:

$$
\begin{aligned}
\boldsymbol{z} &\sim p(\boldsymbol{z}|g_{\theta,\boldsymbol{z}} > f^*, D_n^{\mathcal{X}}) \qquad (4)\\
\boldsymbol{x} &= h_\theta(\boldsymbol{z}),
\end{aligned}
$$

where $g_{\theta,\boldsymbol{z}}|D_n^{\mathcal{X}}$ is the random variable following the GP posterior predictive distribution for the objective value at the decoded structure $\boldsymbol{x} = h_\theta(\boldsymbol{z})$, i.e., $g_{\theta,\boldsymbol{z}} = f(h_\theta(\boldsymbol{z}))$.

By decomposing (4) via Bayes' rule as

$$p(\boldsymbol{z}|g_{\theta,\boldsymbol{z}} > f^*, D_n^{\mathcal{X}}) \propto p(g_{\theta,\boldsymbol{z}} > f^* \mid D_n^{\mathcal{X}}) \times p(\boldsymbol{z}), \quad (5)$$

we can now see that (4) corresponds to sampling from the posterior distribution induced by the VAE's Gaussian prior $p(\boldsymbol{z})$ and a likelihood $p(g_{\theta,\boldsymbol{z}} > f^* \mid D_n^{\mathcal{X}})$, which resembles the PI acquisition function (and so can be calculated easily in closed-form under the GP). Hence, our approximation of COWBOYS reduces to a classical form for which many well-established sampling methods are available.

## 5.2. Preconditioned Crank-Nicolson MCMC

For our experiments, we sample from (4) using a modification of the Preconditioned Crank-Nicolson (PCN) algorithm (Cotter et al., 2013). PCN is well-suited to sampling from medium- to high-dimensional Gaussian priors, as it concentrates on the annulus of high-prior probability (as discussed in Section 3.3) rather than attempting to explore the entire parameter space. This contrasts with more conventional Markov Chain Monte-Carlo (MCMC) algorithms

(e.g., random-walk methods), which rapidly deteriorate in effectiveness as dimensionality increases (Hairer et al., 2014) — an observation closely aligned with our critique of existing LSBO strategies.

Exact details of our implementation of PCN are provided in Appendix A. Note that, in order to leverage parallel hardware, we introduced a minor alteration that transforms PCN into a non-reversible MCMC algorithm, thereby forfeiting the rigorous sample quality guarantees derived by the computational statistics community. However, in the context of COWBOYS, where our ultimate goal is Bayesian optimisation, our empirical findings suggest that producing a set of approximate samples is sufficient.

### 5.3. Computational Complexity

We now analyse the computational overhead incurred by COWBOYS and standard LSBO in non-batch setting. Ignoring the shared computational cost of fitting GP surrogate models, the $n^{th}$ step of a COWBOYS algorithm using PCN with $C$ MCMC chains and $S$ MCMC steps has complexity $O(CS(N^2 + V))$, corresponding to the need to make a GP prediction $O(N^2)$ (assuming a cached Gram matrix inverse) and VAE decoding ( at cost $V$) for each point considered by PCN. In contrast, standard LSBO incurs $O(AN^2 + V)$, where $A$ are the number of evaluations required to maximise its acquisition function. We set up our experiments via choices of $C$ and $S$ so that GP costs are equivalent ($C \times S \approx A$). We stress that although COWBOYS requires more VAE evaluations than standard LSBO, advanced LSBO approaches that fine-tune the VAE on incoming data also require large numbers of additional VAE evaluations.

## 6. Baselines & Related Works

**Improving "alignment"**. The primary recent focus of the LSBO literature has been on allowing VAEs to be fine-tuned during optimisation, encouraging the alignment of latent spaces with respect to the optimisation objective, with Eissman et al. (2018), Tripp et al. (2020), Moriconi et al. (2020), Grosnit et al. (2021), Maus et al. (2022a), and Chen et al. (2024) proposing methods for fine-tuning the latent space mappings — using additional supervised neural networks, re-weighted losses, or performing joint inference over the surrogate model/mapping. Most recently, Lee et al. (2023) fine-tune VAEs with the explicit goal of ensuring the optimisation objective is smooth (in a Lipschitz sense) across the latent space — an approach extended further in Chu et al. (2024) to also address the additional alignment issue arising from stochasticity in the VAE's decoder. Here, LSBO is recast as an inverse problem, where they use optimisation identify the latent codes most likely to decode to any suggested structures before fitting surrogate models.

**Encouraging valid decodings**. Custom VAE decoders have been proposed that can ensure the satisfaction of string-based (Kusner et al., 2017) or graph-based constraints (Jin et al., 2018). Alternatively, the suggestion of valid structures, can be encouraged by using an additional model that predicts the validity of proposed structures to help triage the query points suggested by BO (Griffiths & Hernández-Lobato, 2020) or by avoiding areas of the latent space where the decoder's epistemic uncertainty is high (Notin et al., 2021).

**Additional improvements for LSBO**. Extensions support high-dimensional latent spaces (Maus et al., 2022a), multi-objective (Stanton et al., 2022) and batch optimisation (Maus et al., 2022b), as well as a method to include structure-level knowledge into the surrogate model alongside latent representations (Deshwal & Doppa, 2021). Recent work of (Ramchandran et al., 2025) proposed the use of GP VAEs that allows the inclusion of auxiliary variables to help learn useful latent spaces.

## 7. Results

We compare the performance of COWBOYS over multiple VAES, against 24 unique algorithms, and on 16 open-source benchmark problems for molecular design. Our experiments replicate the exact experimental setups of Chu et al. (2024), González-Duque et al. (2024), and Maus et al. (2024); results are shown in Figure 3, Table 1, and Figure 4, respectively. See (Brown et al., 2019) for the scientific motivation and clear practical details behind these benchmarks.

**General details.** Our code is integrated into the benchmarking suite of González-Duque et al. (2024) and is available at *https://github.com/henrymoss/ROTLSC*. We build on BoTorch (Balandat et al., 2020), GPyTorch (Gardner et al., 2018), and lastly GAUCHE (Griffiths et al., 2024) for its Tanimoto and molecule utilities, adopting all default settings for model optimisation and kernel hyperparameters. Baseline methods are drawn from their respective open-source implementations. For COWBOYS's MCMC-based sampling strategy (our modified PCN, described in Appendix A), we run 100 MCMC steps over a single chain when the batch size is one ($B = 1$), and 50 chains when $B > 5$. This matches the $5,000$ acquisition function evaluation budget typically used by TURBO (Eriksson et al., 2019), a key component of all other high-performance LSBO routines. Appendix B examines COWBOYS' robustness to these MCMC settings and highlights the combined impact of its two primary components by applying COWBOYS-style sampling but with a latent-space GP model.

### 7.1. Comparison with State of the Art LSBO Methods

**Baselines**. Following Chu et al. (2024), we consider **LOL-BO** (Maus et al., 2022a), **CoBO** (Lee et al., 2024), **W-LBO**

(Tripp et al., 2020), **InvBO** (Chu et al., 2024), and **PG-LBO** (Chen et al., 2024), all of which require fine-tuning the VAE during optimisation. We also evaluate the standard **LSBO** method (Algorithm 1) and its trust region variant **TURBO-L** (Eriksson et al., 2019). As a reference point, we include **Graph-GA** (Jensen, 2019), a BO-free baseline that uses a widely adopted graph-based genetic algorithm. For a more detailed discussion of these baselines, see Chu et al. (2024). Note that we used a pre-trained VAE provided via direct correspondence from the authors of (Maus et al., 2022a), as the one provided in their open-source implementation was not sufficient to recreate their results.

**Results.** As shown in Figure 3, COWBOYS achieves marked improvements in sample efficiency over existing LSBO methods in the low-data regime, outperforming even those permitted to fine-tune their VAE. We replicate the experimental setup from Maus et al. (2022a;b); Lee et al. (2024); Chu et al. (2024), focusing on lower-budget problems (COWBOYS cannot currently handle evaluation budgets of 80,000 due to a current lack of availability of scalable GP models for discrete data; see Section 8). Across six tasks introduced by Brown et al. (2019), we run 100 BO steps with a batch size $B = 5$, starting from 100 representative molecules as defined by Lee et al. (2024), a scenario designed to mimic real-world lead optimisation. These experiments employ the SELFIES VAE of Maus et al. (2022a), which encodes and decodes molecules into a 256-dimensional latent space using six transformer layers (see Maus et al., 2022a, for details).

### 7.2. High-dimensional BO Benchmark

**Baselines.** Following the setup in González-Duque et al. (2024), we compare COWBOYS to three categories of methods: (i) non-BO evolutionary optimisers applied to the VAE latent space: **hill-climbing**, **CMAES**, and a Genetic Algorithm (**GA**), (ii) high-dimensional BO methods also operating in the VAE latent space: **RandomLineBO** (Kirschner et al., 2019) and **TURBO** (Eriksson et al., 2019), and (iii) BO methods that optimise directly in the discrete (molecular) space: **Bounce** (Papenmeier et al., 2023) and **ProbRep** (Daulton et al., 2022). For a more detailed discussion of these baselines, see the online documentation in González-Duque et al. (2024).

**Results.** Table 1 shows that COWBOYS consistently outperforms all compared methods on the high-dimensional discrete sequence optimisation tasks provided by González-Duque et al. (2024), (based on the PMO benchmark of Gao et al. (2022)). All algorithms start from an initial design of 10 molecules and run for 300 BO steps with a batch size of $B = 1$. Because the original VAE used by González-Duque et al. (2024) had relatively poor reconstruction accuracy, we reran the entire benchmark (and all baselines) with an

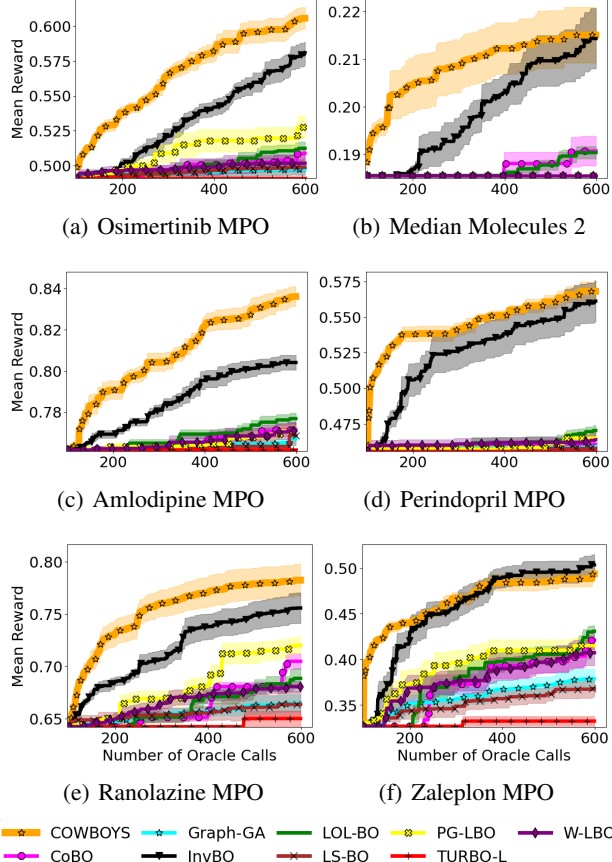

(a) Osimertinib MPO

(b) Median Molecules 2

(c) Amlodipine MPO

(d) Perindopril MPO

(e) Ranolazine MPO

(f) Zaleplon MPO

COWBOYS Graph-GA LOL-BO PG-LBO W-LBO
CoBO InvBO LS-BO TURBO-L

*Figure 3.* Average performance ($\pm$ standard error) of COWBOYS over 10 runs on problems considered by (Chu et al., 2024). COWBOYS achieves a substantial improvement in sample efficiency over all existing LSBO methods (including those able to fine-tune VAEs to incoming data) in this low data regime, with only the recently proposed InvBO sometimes matching its performance. We stress that COWBOYS does not fine-tune its VAE during optimisation, rather just uses it more efficiently.

updated, more expressive, 128-dimensional fully-connected VAE using the latest release of the benchmarking suite.

## 7.3. Comparison with Traditional LSBO

**Baselines**. In our final set of experiments, we restrict our focus to methods that do not fine-tune their VAE. Specifically, we run COWBOYS and LSBO/TURBO-L with exact Gaussian Processes for 100 steps of $B = 20$ evaluations, a limit imposed by the computational demands of exact GP inference. We also include scalable variants—**EULBO EI** and **EULBO KG** (Maus et al., 2024), **Vanilla BO** (Hvarfner et al., 2024), and **IPA** (Moss et al., 2022; 2023)—that rely on Sparse Variational Gaussian Processes (SVGP) (Hensman et al., 2013) to extend LSBO to 4,000 steps. This allows us to examine whether significantly larger optimisation budgets enable these methods to match COWBOYS' performance. Extending COWBOYS to accommodate large-budget optimisation is an intriguing avenue for future work (Section 8). Unlike above, here we used the exact pre-trained VAE provided the open-source implementation of (Maus et al., 2022a) for all approaches.

**Results**. As shown in Figure 4, COWBOYS achieves substantially better performance than other LSBO methods that also do not fine-tune the VAE. Even when given a two orders-of-magnitude increase in optimisation budget, these baselines do not catch up to COWBOYS, highlighting the value of our proposed method of combining VAEs and GPs for Bayesian optimisation. These experiments replicate those in Maus et al. (2024), who perform LSBO using the latent space of a SELFIES VAE (see Section 7.1), focusing on a subset of the molecular design benchmarks discussed earlier.

## 8. Discussion and Limitations

COWBOYS leverages a simple Bayesian updating mechanism that links separately trained VAEs and GPs through a sampling strategy progressively biased toward promising structures. Despite its simplicity, this sampling-based approach directly addresses LSBO's two primary challenges. First, COWBOYS *does not* model the objective in the VAE latent space; instead, it retains the GP in the original, structured representation of the data, allowing the use of specialized kernels. Second, COWBOYS *does not* employ the latent space for optimisation; instead, it naturally restricts its focus to regions that are both plausible under the VAE prior and potentially high-performing under the GP. We demonstrate that COWBOYS offers a substantial boost in sample efficiency on low-budget molecular design benchmarks.

**Extensions into new application areas**. By working in the raw structure space, our approach naturally supports specialized kernels tailored to particular domains—an area

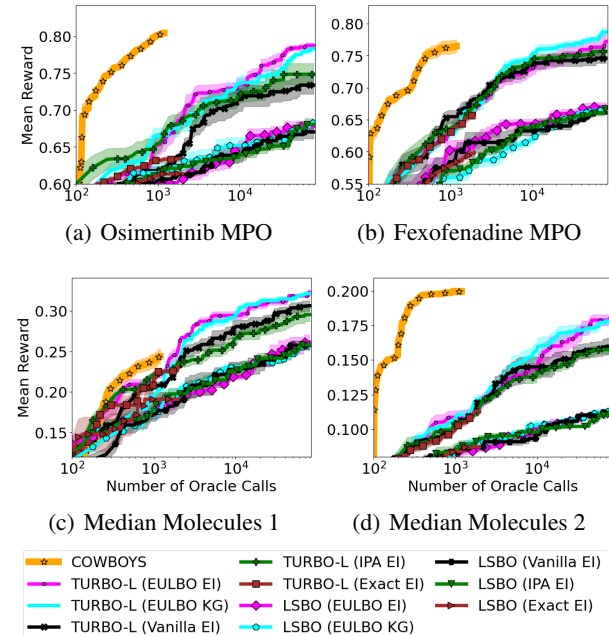

(a) Osimertinib MPO  (b) Fexofenadine MPO

(c) Median Molecules 1  (d) Median Molecules 2

COWBOYS
TURBO-L (EULBO EI)
TURBO-L (EULBO KG)
TURBO-L (Vanilla EI)
TURBO-L (IPA EI)
TURBO-L (Exact EI)
LSBO (EULBO EI)
LSBO (EULBO KG)
LSBO (Vanilla EI)
LSBO (IPA EI)
LSBO (Exact EI)

*Figure 4.* Average performance ($\pm$ standard error) over 20 repetitions with an log-scaled x-axis, demonstrating that, among LSBO methods that cannot fine-tune their latent space, COWBOYS provides significant improvemnt in efficiency.

where there is a rich, yet underused, literature. For instance, the Tanimoto kernel, originally proposed by chemists, encodes prior knowledge of what sort of molecular attributes are important in a way that can be especially powerful in low-data regimes. Similar structural kernels exist for various structured objects and, via COWBOYS, could now be used for molecular graph optimisation with graph kernels (Schraudolph et al., 2010), engineering design with 3d mesh kernels (Perez et al., 2024), optimising computer code via tree kernels (Beck et al., 2015), or protein design with new protein kernels (Groth et al., 2024). By enabling the direct use of such kernels, COWBOYS allows practitioners to incorporate a wealth of domain-specific prior knowledge. We hope our work helps revitalize interest in harnessing these powerful kernels across a range of complex design problems and spurs further synergy between rich generative models and carefully structured discriminative models.

**Methodological extensions**. Extending COWBOYS to the important task of large-scale molecular optimisation raises several exciting avenues for future research. First, we need to develop methods for fitting sparse GPs to discrete molecular data—an inherently more challenging task— where inducing points are more difficult to optimise, perhaps by applying techniques from Burt et al. (2019) or Chang et al. (2022). Second, integrating COWBOYS with LSBO techniques that fine-tune their VAEs during optimisation may become essential for performance once experimental bud-

| Objective function | COWBOYS | HillClimbing | CMAES | GA | RandomLineBO | TURBO | Bounce | ProbRep |
|---|---|---|---|---|---|---|---|---|
| albuterol_similarity | 0.472±0.08 | 0.487±0.06 | 0.453±0.04 | 0.356±0.05 | 0.454±0.06 | 0.456±0.04 | 0.16 ± 0.01 | 0.21 ± 0.03 |
| amlodipine_mpo | 0.477±0.04 | 0.449±0.02 | 0.458±0.02 | 0.440±0.01 | 0.453±0.02 | 0.444±0.02 | 0.00 ± 0.00 | 0.00 ± 0.00 |
| celecoxib_rediscovery | 0.217±0.02 | 0.202±0.01 | 0.213±0.02 | 0.202±0.01 | 0.204±0.01 | 0.202±0.01 | 0.02 ± 0.01 | 0.02 ± 0.00 |
| deco_hop | 0.570±0.01 | 0.562±0.01 | 0.563±0.00 | 0.562±0.01 | 0.564±0.01 | 0.562±0.01 | 0.50 ± 0.00 | 0.51 ± 0.00 |
| drd2_docking | 0.342±0.17 | 0.097±0.09 | 0.087±0.09 | 0.076±0.10 | 0.346±0.37 | 0.170±0.10 | 0.01 ± 0.00 | 0.01 ± 0.00 |
| fexofenadine_mpo | 0.682±0.02 | 0.644±0.04 | 0.652±0.05 | 0.578±0.06 | 0.669±0.03 | 0.632±0.06 | 0.13 ± 0.13 | 0.20 ± 0.08 |
| gsk3_beta | 0.368±0.03 | 0.252±0.05 | 0.321±0.11 | 0.241±0.04 | 0.170±0.05 | 0.302±0.05 | 0.09 ± 0.08 | 0.12 ± 0.02 |
| isomer_c7h8n2o2 | 1.000±0.00 | 0.788±0.19 | 0.872±0.07 | 0.812±0.10 | 0.864±0.15 | 0.880±0.07 | 0.11 ± 0.09 | 0.24 ± 0.11 |
| isomer_c9h10n2o2pf2cl | 0.690±0.08 | 0.600±0.12 | 0.567±0.11 | 0.639±0.07 | 0.654±0.12 | 0.618±0.24 | 0.01 ± 0.01 | 0.06 ± 0.03 |
| jnk3 | 0.138±0.05 | 0.122±0.04 | 0.136±0.02 | 0.130±0.04 | 0.126±0.04 | 0.092±0.03 | 0.05 ± 0.04 | 0.06 ± 0.01 |
| median_1 | 0.198±0.02 | 0.162±0.01 | 0.187±0.03 | 0.137±0.02 | 0.174±0.03 | 0.154±0.04 | 0.03 ± 0.01 | 0.02 ± 0.00 |
| median_2 | 0.162±0.01 | 0.146±0.01 | 0.149±0.01 | 0.145±0.01 | 0.145±0.01 | 0.145±0.01 | 0.01 ± 0.00 | 0.01 ± 0.00 |
| mestranol_similarity | 0.384±0.01 | 0.348±0.02 | 0.362±0.02 | 0.266±0.03 | 0.353±0.01 | 0.378±0.07 | 0.01 ± 0.00 | 0.02 ± 0.00 |
| osimetrinib_mpo | 0.722±0.03 | 0.718±0.02 | 0.711±0.05 | 0.707±0.04 | 0.714±0.03 | 0.711±0.03 | 0.30 ± 0.31 | 0.59 ± 0.04 |
| perindopril_mpo | 0.382±0.04 | 0.326±0.11 | 0.335±0.11 | 0.334±0.12 | 0.337±0.11 | 0.362±0.06 | 0.00 ± 0.00 | 0.00 ± 0.00 |
| ranolazine_mpo | 0.648±0.04 | 0.609±0.02 | 0.625±0.08 | 0.484±0.06 | 0.632±0.03 | 0.597±0.04 | 0.00 ± 0.00 | 0.11 ± 0.02 |
| rdkit_logp | 10.678±0.99 | 11.383±6.12 | 14.474±7.41 | 9.307±1.67 | 19.484±5.92 | 9.691±0.71 | 3.12 ± 1.20 | 5.49 ± 3.01 |
| rdkit_qed | 0.909±0.03 | 0.902±0.03 | 0.912±0.03 | 0.902±0.03 | 0.906±0.04 | 0.903±0.04 | 0.52 ± 0.09 | 0.60 ± 0.05 |
| sa_tdc | 7.909±0.07 | 7.920±0.06 | 7.516±0.55 | 8.878±0.16 | 7.877±0.11 | 7.952±0.05 | 8.36 ± 0.46 | 8.59 ± 0.13 |
| scaffold_hop | 0.435±0.01 | 0.427±0.01 | 0.430±0.01 | 0.427±0.01 | 0.427±0.01 | 0.429±0.01 | 0.34 ± 0.01 | 0.34 ± 0.00 |
| sitagliptin_mpo | 0.259±0.07 | 0.151±0.13 | 0.237±0.12 | 0.154±0.09 | 0.233±0.12 | 0.163±0.12 | 0.00 ± 0.00 | 0.00 ± 0.00 |
| thiothixene_rediscovery | 0.247±0.03 | 0.224±0.02 | 0.224±0.02 | 0.224±0.02 | 0.226±0.02 | 0.224±0.02 | 0.02 ± 0.01 | 0.03 ± 0.01 |
| troglitazone_rediscovery | 0.205±0.02 | 0.187±0.02 | 0.203±0.03 | 0.187±0.02 | 0.188±0.02 | 0.195±0.01 | 0.02 ± 0.01 | 0.02 ± 0.00 |
| valsartan_smarts | 0.000±0.00 | 0.000±0.00 | 0.000±0.00 | 0.000±0.00 | 0.000±0.00 | 0.000±0.00 | 0.00 ± 0.00 | 0.00 ± 0.00 |
| zaleplon_mpo | 0.379±0.05 | 0.337±0.08 | 0.293±0.12 | 0.297±0.11 | 0.344±0.08 | 0.297±0.11 | 0.00 ± 0.00 | 0.00 ± 0.00 |

*Table 1.* Average performance ($\pm$ s.d.) over 5 repetitions of COWBOYS on the discrete BO benchmarking suite of (González-Duque et al., 2024). We stress best average scores achieved after 300 evaluations (dark) and scores within a single standard deviation of best (light).

gets grows and VAE adaptation becomes more feasible. Finally, more sophisticated sampling algorithms could further improve MCMC efficiency (as is important for controlling the cost of COWBOYS under expensive VAE architectures) or allow us to exploit the full stochastic nature of the decoder, rather than restricting the search to the most likely decoding. Also, note that the condition in 3 does resemble the well-known Probability of Improvement (PI) acquisition function. While PI is intuitive, it is known to be suboptimal and can exhibit pathological behaviour and so adapting more sophisticated acquisition schemes into a COWBOYS framework is an exciting direction for future research, e.g. light-weight information-theoretic schemes for multi-fidelity (Moss et al., 2020b), multi-objective (Qing et al., 2023b) and tackling more ambitious problems like quantile optimisation (Picheny et al., 2022).

## Acknowledgments

This research was supported by AstraZeneca, Schmidt Sciences, and Lancaster University's Mathematics for AI in Real-world Systems E3 grant.

## Impact Statement

Our work introduces an efficient framework for optimising over discrete sequences, a problem of growing interest in machine learning. Real-world benefits in molecule design include significantly reducing wet-lab resources. However, the same method could be misused in non-therapeutic applications. Thus, while optimisation is automatic, humans must decide how these results are applied, following appropriate ethical guidelines.

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

# A. COWBOYS' sampler details

We now provide full details for our PCN MCMC sampler (Cotter et al., 2013) as Algorithms 3 and 4 which we use to get a set of samples from COWBOYS sampling objective (4). Note that we keep track of the latent value that decoded to give the best structure so far $z_{best}$ and use this to start our chains. We also employ an adaptive choice of PCN's scale parameter $\beta$ following Andrieu & Thoms (2008). Often, especially early in the optimisation before the region satisfying likely to satisfy $f(x) > f^*$ narrows, our PCN sampler can return a larger number of samples than our desired batch size. In this case, we select the $B$ samples that achieve the highest expected utility under greedy maximisation of the batch Expected Improvement (qEI) acquisition function provided in BoTorch (Balandat et al., 2020). If very large batches are required, then a subset could be chosen using Thompson following Vakili et al. (2021). Although this subset selection strategy breaks MCMC's detailed balance, it ensures that we return a diverse, high-value subset of samples to COWBOYS – samples that are likely to be approximate anyway due to relatively small number of MCMC iterations and lack of burn-in phase. Note that in practical implementations, our PCN (Algorithm 4) is trivially parallelised by running all chains present in the outer loop at once.

---

**Algorithm 3** COWBOYS' MCMC Sampler

---

**Input:** Batch size $B$, number chains $C$, number steps $S$, threshold $f^*$, starting latent code $z_{best}$
$Z_{samples} \leftarrow \text{PCN}(C, S, f^*, z_{best})$           ◁ Algorithm 4
**while** $|Z_{samples}| < B$ **do**           ◁ Keep sampling if needed
    $Z_{samples} \leftarrow Z_{samples} \bigcup \text{PCN}(C, S, f^*, z_{best})$           ◁ Algorithm 4 again
**end while**
**if** $|Z_{samples}| > B$ **then**
    $Z_{samples} \leftarrow \underset{\substack{Z \subseteq Z_{samples} \\ :|Z|=B}}{\text{argmax}} \text{qEI}(Z)$           ◁ choose $B$ best according to batch utility
**end if**
**return** $Z_{chosen}$

---

---

**Algorithm 4** PCN

---

**Input:** number chains $C$, number steps $S$, best score so far $f^*$, starting latent code $z_{best}$
$Z_{samples} \leftarrow \emptyset$
**for** $c \in \{1, .., C\}$ **do**
    $\beta \leftarrow 0.1$           ◁ Initialise step size
    $z_{\text{current}} \leftarrow z_{\text{best}}$           ◁ Initialise new chain at best so far
    **for** $n \in \{1, .., S\}$ **do**           ◁ Do MCMC step
        $z_{\text{proposal}} \sim \mathcal{N}_d(\sqrt{1 - \beta^2} z_{\text{current}}, \beta^2 I_d)$           ◁ Sample from PCN proposal density
        $\alpha \leftarrow \min\left(1, \frac{p\left(g_{\theta, z_{\text{proposal}}} > f^* | D_n^{\mathcal{X}}\right) \times p(z_{\text{proposal}})}{p\left(g_{\theta, z_{\text{current}}} > f^* | D_n^{\mathcal{X}}\right) \times p(z_{\text{current}})}\right)$ ◁ Calculate acceptance probability using prior and likelihood from (5)
        $u \sim U[0, 1]$
        **if** $\alpha > u$ **then**           ◁ Stochastic acceptance step
          $z_{\text{current}} \leftarrow z_{\text{proposal}}$
          $Z_{samples} \cup \{z_{\text{current}}\}$           ◁ Store sample
        **end if**
        $\beta \leftarrow \beta + 0.1(\alpha - 0.243)$           ◁ Adaptive MCMC update
    **end for**
**end for**
**return** $Z_{samples}$           ◁ return all accepted samples

---

| Oracle | COWBOYS 10 chains 100 steps | COWBOYS 1 chains 100 steps | COWBOYS 10 chains 50 steps | COWBOYS 1 chains 50 steps | COWBOYS 1 chains 10 steps | COWBOYS with Latent GP 10 chains 100 steps |
|---|---|---|---|---|---|---|
| albuterol_similarity | 0.454+/-0.06 | 0.472+/-0.08 | 0.450+/-0.06 | 0.435+/-0.05 | 0.464+/-0.07 | 0.459+/-0.09 |
| amlodipine_mpo | 0.461+/-0.02 | 0.477+/-0.04 | 0.471+/-0.02 | 0.450+/-0.01 | 0.461+/-0.02 | 0.457+/-0.01 |
| celecoxib_rediscovery | 0.264+/-0.04 | 0.217+/-0.02 | 0.240+/-0.03 | 0.224+/-0.02 | 0.240+/-0.04 | 0.202+/-0.00 |
| deco_hop | 0.569+/-0.01 | 0.570+/-0.01 | 0.572+/-0.02 | 0.562+/-0.01 | 0.568+/-0.01 | 0.562+/-0.01 |
| drd2_docking | 0.214+/-0.23 | 0.342+/-0.17 | 0.325+/-0.17 | 0.217+/-0.21 | 0.266+/-0.08 | 0.241+/-0.26 |
| fexofenadine_mpo | 0.671+/-0.02 | 0.682+/-0.02 | 0.681+/-0.02 | 0.677+/-0.01 | 0.698+/-0.02 | 0.674+/-0.03 |
| gsk3_beta | 0.410+/-0.02 | 0.368+/-0.03 | 0.358+/-0.06 | 0.354+/-0.10 | 0.322+/-0.05 | 0.290+/-0.03 |
| isomer_c7h8n2o2 | 0.934+/-0.05 | 1.000+/-0.00 | 0.889+/-0.06 | 0.976+/-0.05 | 0.921+/-0.06 | 0.880+/-0.07 |
| isomer_c9h10n2o2pf2cl | 0.634+/-0.06 | 0.690+/-0.08 | 0.653+/-0.05 | 0.634+/-0.08 | 0.649+/-0.04 | 0.622+/-0.05 |
| jnk3 | 0.138+/-0.01 | 0.138+/-0.05 | 0.124+/-0.02 | 0.146+/-0.02 | 0.144+/-0.01 | 0.134+/-0.03 |
| median_1 | 0.211+/-0.02 | 0.198+/-0.02 | 0.218+/-0.02 | 0.198+/-0.03 | 0.175+/-0.02 | 0.170+/-0.03 |
| median_2 | 0.156+/-0.01 | 0.162+/-0.01 | 0.150+/-0.01 | 0.155+/-0.01 | 0.158+/-0.01 | 0.145+/-0.01 |
| mestranol_similarity | 0.423+/-0.04 | 0.384+/-0.01 | 0.458+/-0.05 | 0.380+/-0.03 | 0.388+/-0.03 | 0.362+/-0.04 |
| osimetrinib_mpo | 0.749+/-0.02 | 0.722+/-0.03 | 0.731+/-0.01 | 0.733+/-0.02 | 0.754+/-0.02 | 0.721+/-0.02 |
| perindopril_mpo | 0.390+/-0.03 | 0.382+/-0.04 | 0.388+/-0.03 | 0.376+/-0.04 | 0.399+/-0.03 | 0.357+/-0.06 |
| ranolazine_mpo | 0.652+/-0.03 | 0.648+/-0.04 | 0.643+/-0.02 | 0.638+/-0.03 | 0.628+/-0.04 | 0.601+/-0.02 |
| rdkit_logp | 9.733+/-0.96 | 10.678+/-0.99 | 10.508+/-0.57 | 10.690+/-0.69 | 9.517+/-0.53 | 9.406+/-1.34 |
| rdkit_qed | 0.917+/-0.02 | 0.909+/-0.03 | 0.926+/-0.01 | 0.915+/-0.02 | 0.906+/-0.03 | 0.917+/-0.02 |
| sa_tdc | 8.047+/-0.15 | 7.909+/-0.07 | 8.012+/-0.13 | 7.982+/-0.02 | 7.945+/-0.03 | 7.798+/-0.13 |
| scaffold_hop | 0.451+/-0.02 | 0.435+/-0.01 | 0.441+/-0.02 | 0.444+/-0.02 | 0.440+/-0.01 | 0.427+/-0.01 |
| sitagliptin_mpo | 0.388+/-0.05 | 0.259+/-0.07 | 0.277+/-0.06 | 0.246+/-0.03 | 0.247+/-0.06 | 0.276+/-0.09 |
| thiothixene_rediscovery | 0.243+/-0.02 | 0.247+/-0.03 | 0.244+/-0.02 | 0.238+/-0.01 | 0.247+/-0.02 | 0.225+/-0.01 |
| troglitazone_rediscovery | 0.197+/-0.01 | 0.205+/-0.02 | 0.208+/-0.02 | 0.214+/-0.02 | 0.203+/-0.02 | 0.192+/-0.01 |
| valsartan_smarts | 0.000+/-0.00 | 0.000+/-0.00 | 0.000+/-0.00 | 0.000+/-0.00 | 0.000+/-0.00 | 0.000+/-0.00 |
| zaleplon_mpo | 0.382+/-0.03 | 0.379+/-0.05 | 0.396+/-0.01 | 0.346+/-0.05 | 0.357+/-0.04 | 0.340+/-0.07 |

*Table 2.* Average performance ($\pm$ s.d.) over 5 repetitions of COWBOYS on the discrete BO benchmarking suite of (González-Duque et al., 2024). We stress best average scores achieved after 300 evaluations (dark) and scores within a single standard deviation of best (light). All runs of COWBOYS achieve roughly comparable results except the two on the far right, i.e, using too few MCMC chains/iterations or modifying COWBOYS to use a latent space GP, respectively.

## B. COWBOYS' Ablation Study

Table 2 examines COWBOYS' robustness to different configurations of its PCN MCMC sampler across the benchmark problems of González-Duque et al. (2024), demonstrating that COWBOYS is not sensitive to the number of MCMC chains or the number of MCMC steps, unless they are set to be very small. Of course, no single parameter choice can yield uniformly optimal performance across different objective functions, owing to varying degrees of model mismatch between our GP and each specific problem objective and varying degrees of difficulty/locality of the optimisation problems. The final column of the table highlights the combined impact of COWBOYS' two primary components (its MCMC sampling, and its modelling in structure space) by demonstrating that if we apply COWBOYS-style sampling but with a latent-space GP model, we lose performance. More precisely, rather than fitting a Tanimoto GP in the structure space, we instead follow an approach closer to LSBO, and fit a latent space GP, i.e., we replace the structured GP model $\hat{f}$ (recall $\hat{f}(h_\theta(z))$) with a direct GP model $\tilde{g}(z)$, but otherwise proceed as in COWBOYS.

In order to diagnose if our deterministic decoder approximation is limiting the flexibility of our proposed molecules, We also ran the MCMC as described (to get a chosen **z**) but then sampled from the full stochastic decoder to get the chosen molecule. This lead to a significant drop in performance for 10 of the 25 tasks (and returning statistically similar scores for the other tasks), which we hypothesise is due to the fact that the resulting "sampled" molecules are now not the same as the ones passed to the GP to see if the considered latent code will yield an improvement in score – i.e. we return to a problem similar to that of the alignment issues that we discuss in Section 3.2.

Finally, we also investigated the robustness of COWBOYS against degraded surrogate mode performance (to mimic settings where it may be hard to propose a structure-space kernel). Here, we deliberately impaired the Tanimoto GP surrogate by randomly swapping ones to zeros in molecular fingerprints with varying probabilities. This experiment corresponds to adding significant input noise to the fingerprints by randomly removing the count of particular (hashed) molecular substructures. Our findings indicated that COWBOYS maintains robust performance except under the most severe perturbations. In summary, perturbing Tanimoto entries with probability 0.01, 0.1, and 0.5 lead to a statistically significant drop in the best molecule found across 0, 3, and 8 tasks from the 25 molecular optimization tasks, respectively.

