# OpenReview forum: "Return of the Latent Space COWBOYS: Re-thinking the use of VAEs for Bayesian Optimisation of Structured Spaces"
_ICML.cc/2025/Conference — ICML 2025 spotlightposter_

### Official Review · Reviewer_vn27 · 2025-03-06

**Overall Recommendation:** 4

**Summary:**

In the proposed manuscript, the authors present a novel approach for Bayesian optimization within the latent space of a VAE applied to structured domains like molecular design. They aim to address multiple drawbacks of conventional methods that often result in suboptimal performance when the latent space does not align with the specific task. To address these issues, the authors introduce a decoupled framework where the generative model and a GP surrogate are trained independently and later integrated through a principled Bayesian update rule. This approach allows the VAE to specialize in generating candidate structures while the GP excels in predictive modeling, thereby improving the identification of promising candidates under limited evaluation budgets.

**Claims And Evidence:**

The paper claims that while BO in the latent space of a VAE is a powerful framework for tackling complex optimisation tasks,  existing methods have significant drawbacks, including suboptimal surrogate modeling in the latent space and inherent search space challenges.  The paper provides a thorough discussion of these issues, offering both descriptive analysis and empirical evidence. Moreover, the proposed decoupled approach appears to address these shortcomings effectively, potentially serving as a remedy for the identified limitations.

**Essential References Not Discussed:**

To the best of my knowledge, the authors have included the key literature in the field, particularly regarding LSBO approaches.

**Experimental Designs Or Analyses:**

The experimental design seems in line with other literature in the field and appears to be well-integrated within a benchmark suite. The approach also sounds plausible based on the information presented.

**Methods And Evaluation Criteria:**

The proposed method decouples the search space from the latent space and further uses a Preconditioned Crank-Nicolson algorithm (which concentrates on high-prior probability annulus, better than random walk sampling) for sampling.

COWBOYS conducts a number of studies, showing competitive results with state-of-the-art comparison methods, even those that fine-tune the VAE, and with high-dimensional BO benchmarks and traditional LSBO methods. Finally, the authors discuss the need to develop/extend COWBOYS to large-scale molecule optimization tasks, as this can be challenging with the proposed structure-based GP optimization approach.

While the presented approach shows promising results on the current task, the manuscript's impact would be significantly enhanced by testing the method in additional domains, such as protein fitness optimization or similar applications involving other structured spaces.

**Other Comments Or Suggestions:**

line 98 left: expected improvement twice

line 82 right: this should be encoder instead of decoder

line 94 right: “.” missing at the end of the sentence

line 227 left: In algorithm, the arrow extends into the brackets

Line 269 left: “*Note that replacing the VAE’s probabilistic decoding with the most-likely mapping is already a common strategy in many LSBO implementations”* You are citing here a survey paper that does not discuss this in detail, please cite the actual works to make the connection clear for the reader.

**Other Strengths And Weaknesses:**

Is the search space problem also an issue for the initial design of COWBOYS? Essentially, the initial design of Alg. 2 also requires sampling from the VAE’s search space and decoding the samples.

**Questions For Authors:**

See questions asked above.

**Relation To Broader Scientific Literature:**

The authors start by motivating LSBO, which has emerged to be a prominent technique in this field in recent years. Current limitations and shortcomings of LSBO are discussed, and how the community handles some of these (e.g., fine-tuning the VAE, limiting the search space). This is followed by how COWBOYS addresses these issues.

**Theoretical Claims:**

The theoretical claims in the presented work seem plausible and well discussed.

---

> ### Author Rebuttal · Authors · 2025-03-31
>
> Thank you for your thoughtful assessment of our work. We appreciate your recognition of both the descriptive and empirical strengths of our approach. It is encouraging to hear that you find our decoupled strategy promising, and that our competitive results align well with other state-of-the-art methods.
>
> We have addressed all your suggestions thoroughly and have implemented the requested revisions as detailed below. We hope that our efforts to address the comments from you and the other reviewers will be appreciated and could justify increasing your score.
>
> 1) **Generalizability of COWBOYS Beyond Molecular Design and Drug Discovery**. We appreciate your interest in the broader applicability of COWBOYS outside its current focus on molecular design and drug discovery. Our evaluation focuses on these tasks because they remain the most widespread use cases for latent space Bayesian optimization (LSBO), and well-established baseline suites exist to facilitate direct comparison.
> Fitting a Gaussian Process (GP) in the original, potentially high-dimensional and highly structured space can indeed be challenging with standard kernels. However, we see an ability to model directly in structure space as a key advantage of COWBOYS, rather than a limitation. By working in the raw structure space, our approach naturally supports specialized kernels tailored to particular domains—an area where there is a rich, yet underused, literature. For instance, the Tanimoto kernel, originally proposed by chemists, encodes prior knowledge of what sort of molecular attributes are important in a way that can be especially powerful in low-data regimes [1,2]. Similar structural kernels exist for various structured objects and, via COWBOYS, could now be used for molecular graph optimisation with graph kernels [3], engineering design with 3d mesh kernels [4], optimising computer code via tree kernels [5], or protein design with new protein kernels [6].
> In contrast, many current LSBO strategies cannot leverage these specialized kernels because they rely on Euclidean latent spaces. By enabling the direct use of such kernels, COWBOYS allows practitioners to incorporate a wealth of domain-specific prior knowledge. We hope this work helps revitalize interest in harnessing these powerful kernels across a range of complex design problems and spurs further synergy between rich generative models and carefully structured discriminative models
> 2) **We are pleased to hear that our discussion clarified the robustness of COWBOYS with respect to PCN parameters**. However, In the revised manuscript, we have added details on how to interpret the corresponding table, emphasizing that no single parameter choice can yield uniformly optimal performance across different objective functions, owing to varying degrees of model mismatch between our GP and each specific problem objective and varying degrees of difficulty/locality of the optimisation problems.
>
> 3) **“Is the search space problem also an issue for the initial design of COWBOYS”?** In standard LSBO, a specific region of the latent space must be chosen, over which we sample an initial design and then perform BO. We argue in Section 3.3 that this clipping can lead to pathological performance issues. In contrast, COWBOYS just samples from the VAE in the standard way, i.e. we generate a random Gaussian (which has infinite support) and then push it through the decoder. There is no need to specify a search space for COWBOYS. We have greatly increased discussion around this point in the final version.
>
> 4) Also, thanks for the **minor corrections**, they have now been changed in the final version, including adding more four specific references to justify ”replacing the VAE’s probabilistic decoding with the most-likely mapping is already a common strategy”.
>
> [1] Griffiths, Ryan-Rhys, et al. "GAUCHE: a library for Gaussian processes in chemistry." Advances in Neural Information Processing Systems 36 (2023): 76923-76946.
>
> [2] Moss, Henry, et al. "Boss: Bayesian optimization over string spaces." Advances in neural information processing systems 33 (2020): 15476-15486.]
>
> [3] Vishwanathan, S. Vichy N., et al. "Graph kernels." The Journal of Machine Learning Research 11 (2010): 1201-1242.
>
> [4] Perez, Raphaël Carpintero, et al. "Gaussian process regression with Sliced Wasserstein Weisfeiler-Lehman graph kernels." International Conference on Artificial Intelligence and Statistics. PMLR, 2024.
>
> [5] Beck, Daniel, et al. "Learning structural kernels for natural language processing." Transactions of the Association for Computational Linguistics 3 (2015): 461-473.
>
> [6] Groth, Peter Mørch, et al. "Kermut: Composite kernel regression for protein variant effects." Advances in Neural Information Processing Systems 37 (2024): 29514-29565.

---

> > ### Comment · Reviewer_vn27 · 2025-04-03
> >
> > Thank you for the clarification, I raised my recommendation to "Accept".

---

### Official Review · Reviewer_QRHf · 2025-03-11

**Overall Recommendation:** 4

**Summary:**

This paper looks for valid molecules with a high probability of improving a desired property of the molecule. The novelty of this method is that they decouple the two probabilities: the probability of a molecule existing is estimated by VAE which is initially trained/pre-trained, then the probability of a molecule having the desired properties is estimated from a GP which operates on the space of count based vectors of molecular features.

By not fitting a GP in the latent space of the VAE they claim to avoid two pathologies. First,  there can be a lack of smoothness of the objective function in the latent space. Second, because a single point in the VAE latent space maps to several molecules through noise added to the decoder which each have different values for the objective function, a GP in the latent space will inevitably have more noise than could be predicted from a single structure.

The authors suggest candidate molecules using an MCMC sampler using a prior defined by a deterministic decoder and a likelihood given by the probability of improvement compared to the current best observation, as calculated by the GP.

The authors benchmark this method agains various alternative methods on several datasets.

**Claims And Evidence:**

The authors claim that their method outperforms alternatives the low data regime and existing high dimensional discrete sequence which appears to be well supported by the experiments shown here.  They also claim with good support to outperform traditional Latent Space BO even when the traditional methods have much higher experimental budgets.

Two elements which make me less confident on the quality of the method are:

1. The code has been retracted for the review and I do not see any anonymised version to check their implementation. I would have appreciated seeing the code during the review.

2. Use of different comparison methods for different datasets.  I am not familiar with the standard benchmarks in this area and would have appreciated either seeing the same methods applied across all datasets, or a justification for why not. Please see the question below regarding justifying this.

**Essential References Not Discussed:**

I am not aware of any references which should have been discussed here, although I do not claim extensive knowledge of BO applied to molecular design.

**Experimental Designs Or Analyses:**

I do not see any issues with the experimental design carried out here, the experiments and comparisons both seem to be valid and show good results.

**Methods And Evaluation Criteria:**

I thought that the proposed method and evaluation criteria mostly made sense and was well justified.

My one concern was in Section 5.1 where they discard the stochasticity of the decoder. The VAE with a stochastic decoder should provide a probabilistic manifold for the density of possible molecules. The deterministic decoder seems to have a potential issue with either assigning definite existence to impossible molecules or definite existence to optimal molecules. Additionally, as the MCMC sampler was just used to find candidate molecules (as opposed to finding exact posteriors) I do not see the issue with running the MCMC as described but then sampling from the stochastic decoder, which would have potentially avoided these concerns.

I would have appreciated a discussion of potential issues from this, although I appreciate that they do not seem to have been major concerns in practice given the experimental results. Please also see the question regarding this.

**Other Comments Or Suggestions:**

1. As a non-expert in BO in molecular design I would have liked 1-2 sentences describing each of the benchmark datasets and the particular challenge/reason for including each one.

2. The equation on the left hand column of line 267 was unclear if it was using a Dirac Delta function, which is typically not used for categorical variables. I believe that what is being stated in that line is that $\hat p_\theta(x|z)$ is a vector of all zeros except for one element with value one, where the non-zero element corresponds to the maximum likelihood prediction for $p_\theta(x|z)$. See also the question on this.

**Other Strengths And Weaknesses:**

The strengths and weaknesses are addressed in other comments adequately.

**Questions For Authors:**

1. Can you comment on possible problems of using a deterministic decoder, specifically:
     i) is there a risk of the VAE with a deterministic decoder assigning 100% confidence to molecules which in practice do not/ cannot exist?
     ii) Does the deterministic VAE prevent the model from exploring other similar molecules to high performing ones result in missing very good candidate molecules, especially in the final rounds of a BO run in the exploitative stage?

2. Can you clarify what is meant by the equation on the left hand column at line 267?

3. Can you clarify why on different benchmark datasets different comparison methods are used? Do the different approaches not work on the other benchmark datasets, or was the decision for which methods to compare based on another consideration?

**Relation To Broader Scientific Literature:**

I am not familiar with the literature surrounding BO for molecules, but to me the literature review seems extensive.

**Theoretical Claims:**

The authors make no strong theoretical claims.

---

> ### Author Rebuttal · Authors · 2025-03-31
>
> Thank you very much for your thoughtful review and positive feedback regarding our experimental design and validations. We greatly appreciate your recognition of our method's strong performance. We have carefully addressed all of your suggestions and implemented the requested revisions, detailed below. We hope these efforts demonstrate our commitment to improving the manuscript and may justify an increased evaluation score.
>
> 1) **Code Accessibility** In response to your suggestion we have now made some of our code available [here](https://anonymous.4open.science/r/cowboys-AC3E/README.md). The repository allows the recreation of the benchmarking results of Section 7.2. Indeed, incorporating COWBOYS (see [cowboys.py](https://anonymous.4open.science/r/cowboys-AC3E/cowboys.py)) required only minimal additions, emphasizing both its practical usability and potential for broad community adoption. The fully-fledged codebase, allowing COWBOYS to be applied to generic pre-trained VAEs and thus applied to a range of downstream tasks (and recreate the results of Sections 7.1 and 7.2), will be provided upon acceptance as it cannot be shared without violating ICML’s anonymity rules.
>
> 2) **Clarification of Equation (Line 267)** We appreciate you pointing out the ambiguity in the equation on line 267. As requested, we have clarified this equation explicitly in the revised manuscript, clearly defining the delta function as indicating a deterministic decoding strategy that selects only the most-likely decoded molecule from the latent location z. The delta function here means that the probability is 0 unless we are taking the most likely prediction from the decoder (from the chosen latent location z).
>
> 3) **Addressing concerns about discarding the stochasticity of the decoder**. We understand your concerns, as we initially were also worried. We agree that there is indeed a theoretical risk of being unable to generate as wide a range of molecules as the stochastic VAE (although not one we observed in practice). We believe the risks of allocating 100% confidence to implausible molecules, which would arise when decoding areas of the latent space where the decoder is highly uncertain, are strongly mitigated by the fact that we are doing conditional VAE sampling (i.e. staying on the well-supported areas of the latent space) rather than exploring the whole latent space like standard LSBO methods.  Nevertheless, future work will seek more sophisticated sampling strategies to perform COWBOYS-like strategies for generative models (see our Discussion Section).
> We like your idea of running the MCMC as described but then sampling from the decoder in order to properly diagnose if the restrictions of the deterministic decoder are limiting the flexibility of our proposed molecules. We have implemented and ran this suggestion for the high-dimensional BO benchmark of Section 7.2, with the full results added to the ablation study of Appendix B. Unfortunately, this lead to a significant drop in performance for 10 of the 25 tasks (and returning statistically similar scores for the other tasks), which we hypothesise is due to the fact that the resulting “sampled” molecules are now not the same as the ones passed to the GP to see if the considered latent code will yield an improvement in score – i.e. we return to a problem similar to that of the alignment issues that we discuss in Section 3.2. Based on your comments,  we have greatly increased our discussion of our decision to use a deterministic simplification of the decoder, stressing the points above and that (1) the VAE is still initially trained with a stochastic decoder (it is only at inference/BO time that we take the most likely mapping), and (2) we have added four additional specific references of previous work that also using this deterministic simplification.
>
> 4) **Clarifying Differing Comparisons in Experiments**.  The diversity in initial budgets, degrees of parallelism, and subsets of molecular optimization problems stems directly from our intention to replicate and extend the experiments used in influential prior work in LSBO precisely. We believe this breadth provides a robust and comprehensive empirical evaluation of COWBOYS' performance. For peace of mind, please see the comments of the other reviewers, e.g. “Empirical evaluations use the standard benchmarks in the field and replicates and extends the evaluations from previously published papers” and “experiments are conducted thoroughly by evaluating a number of baselines on a good number of test problems”. However, acknowledging your valid concern regarding transparency—especially for readers less familiar with molecular design benchmarks—we have now included an additional section in the appendix. This section explicitly details the structure and interpretation of these benchmark suites, clearly explaining the objectives, settings, and significance of the various optimization tasks we considered.

---

### Official Review · Reviewer_1JvV · 2025-03-12

**Overall Recommendation:** 4

**Summary:**

This manuscript proposes an alternative approach to latent space Bayesian optimization in high-dimensional/structured spaces. Contrary to previous methods that aim to efficiently couple a generative model (typically VAE) and GP surrogate, the proposed method decouples the generative model and GP surrogate.  The proposed method builds on an assumption that a surrogate model can be trained in the original high-dimensional/structured data domain or using features extracted from the data objects in the original domain (here authors consider molecules and features are molecular fingerprints). The trained generative model and GP surrogate are then combined during the optimization step by defining a conditional sampling distribution for the evaluation of next molecules. Results section contains extensive evaluation against most if not all the state-of-the-art methods, and results demonstrate highly competitive performance.

Review update after rebuttal: I am satisfied with authors answers, and will keep my suggestion to accept the manuscript.

**Claims And Evidence:**

The presented result support the claims of highly competitive performance.

**Essential References Not Discussed:**

All relevant background is presented.

**Experimental Designs Or Analyses:**

Empirical evaluations use the standard benchmarks in the field and replicates and extends the evaluations from previously published papers.

**Methods And Evaluation Criteria:**

The proposed method can be considered as a well-designed and novel approach to the high-dimensional/structured BO.  Empirical evaluations follow the standard benchmarks in the field.

**Other Comments Or Suggestions:**

I agree with all central parts of the manuscript. My only slightly critical comment or concerns is that the proposed method assumes that the surrogate model can be trained using the molecular fingerprints as inputs. More generally, the proposed method is applicable whenever a structured GP kernel can be defined, as authors note at the end of page 1 (this limitation could be repeated in Limitations Section at the end).  While this works very well for molecules using fingerprints and Tanimoto kernel, there certainly are other problems where it less clear how a structured kernel should be defined, or whether such exist.  I have two related suggestions: 1) authors could elaborate on when and why this assumption holds (e.g. in Discussion and Limitations Section).  2) authors could carry out an ablation where they study the role and accuracy of the structured surrogate GP.  Authors could e.g. artificially remove (or add noise to) fingerprints such that the surrogate modeling becomes more challenging and less accurate, and evaluate how that affects the overall performance.

Notation: my understanding is that throughout the manuscript $x$ generally denotes a molecule, except in the context of Tanimoto kernel, where $x$ denotes the molecular fingerprint representation. I suggest to use a different symbol around Tanimoto kernel, e.g. $finger(x)$  or something else to clearly note that they are different.

Since your model is build on probability of improvement (PI) acquisition function, it would make sense to use the same acquisition function also for the baseline comparison methods. Perhaps that was mentioned in the text but I do not remember seeing that authors commented on that.

Can you also incorporate a probabilistic graphical model representation of your proposed model (in supplement if not enough space in the main text).

The proposed method can be considered as a conditional generative model for BO. In related works section you could also cite a recent work (Ramchandran et al, ICLR, 2025, https://openreview.net/pdf?id=SIuD7CySb4 ) that has also proposed to use conditional generative model when they attempt to align the latent space for surrogate modeling.

**Other Strengths And Weaknesses:**

As far as I can tell, the proposed method can be considered as a novel approach to BO in structured spaces.

The method is presented clearly and entire manuscript is very well written.

Empirical performance evaluation seems solid, and the results demonstrate strong performance.

**Questions For Authors:**

-

**Relation To Broader Scientific Literature:**

see below

**Theoretical Claims:**

Manuscript does not contain any formal theoretical claims but the probabilistic model construction seems well-designed and valid.

---

> ### Author Rebuttal · Authors · 2025-03-31
>
> Thank you very much for your thoughtful review and valuable suggestions. We sincerely appreciate your recognition of the novelty, clarity, rigorous empirical evaluation, comprehensive background coverage, and overall manuscript quality of our work.
> We have addressed all your suggestions thoroughly and have implemented the requested revisions as detailed below:
> 1) **Generalizability of COWBOYS Beyond Molecular Design and Drug Discovery**: We fully agree with your observation on the importance of clarifying the generalizability of our approach. Following your advice, we have expanded our discussion (now included prominently in the introduction and highlighted in the limitations section) regarding the applicability and significant potential of COWBOYS beyond molecular fingerprint-based problems. Please refer to our detailed response to reviewers vn27 and 1JvV for further context.
> Furthermore, your insightful suggestion about investigating the robustness of COWBOYS against degraded surrogate mode performance (to mimic settings where it may be hard to propose a structure-space kernel) was extremely helpful. As suggested, we have conducted additional experiments on the high-dimensional BO benchmark of Section 7.2, deliberately impairing the Tanimoto GP surrogate by randomly swapping ones to zeros in molecular fingerprints with varying probabilities. This experiment corresponds to  adding significant input noise to the fingerprints by randomly removing the count of particular (hashed) molecular substructures. Our findings indicate that COWBOYS maintains robust performance except under the most severe perturbations and have been added to our ablation studies of Appendix B. In summary, perturbing Tanimoto entries with probability 0.01, 0.1, and 0.5 lead to a statistically significant drop in the best molecule found across 0, 3, and 8 tasks from the 25 molecular optimization tasks, respectively.
> 2) **Small additional Points**. We have clarified our notation by using boldface m for fingerprints to distinctly differentiate from molecules represented by boldface x. Following your recommendation, we have included a graphical model illustrating the dependencies within both COWBOYS and existing LSBO approaches. We appreciate this suggestion as it significantly enhances the manuscript's clarity. We have added the suggested recent reference, highlighting its relevance. The presence of this paper (and also [1] at the same conference in 2 months time), both using extensions of standard LSBO frameworks, underscores the timeliness and importance of our contributions.
>
> Thank you once again for your constructive feedback, which has significantly strengthened our manuscript.
>
> [1] Lee, Seunghun, et al. "Latent Bayesian Optimization via Autoregressive Normalizing Flows." The Thirteenth International Conference on Learning Representations. 2025.

---

### Official Review · Reviewer_dWxR · 2025-03-18

**Overall Recommendation:** 4

**Summary:**

This paper proposes a novel approach to Bayesian Optimization (BO) over structured spaces such as molecular design. Instead of fitting a Gaussian Process (GP) surrogate model within the latent space of a Variational Autoencoder (VAE), which often leads to poor predictive performance, the authors introduce COWBOYS (Categorical Optimisation With Belief Of underlYing Structure). This method decouples the generative model (VAE) from the surrogate (GP), allowing the GP to be trained directly in the structured space rather than the latent space. By leveraging a Bayesian update rule, COWBOYS refines its search iteratively without requiring the complex fine-tuning of VAEs. Experiments show the approach outperforms traditional Latent Space BO (LSBO) methods, demonstrating significant improvements in efficiency for molecular optimization tasks under constrained evaluation budgets.

**Claims And Evidence:**

Overall yes, though with some of my concerns listed below.

**Essential References Not Discussed:**

N/A

**Experimental Designs Or Analyses:**

Overall experiments are conducted thoroughly by evaluating a number of baselines on a good number of test problems.

Ideally I’d like to see more ablation study on COWBOYS’ components (e.g., sensitivity to VAE hyperparameters, impact of optimization budget such as N_init, choice of acqf used in eq 3), as we currently only have a very limited one for the PCN MCMC sampler in Appendix B.

**Methods And Evaluation Criteria:**

Yes, though the evaluation is mostly done on molecular design and drug discovery problems. Although they are the area of application that COWBOYS is designed for, it is unclear whether the claimed superior performance can be extended to other application domains.

One of the issues of the proposed method is that it still requires fitting a GP (or other probabilistic model) on the original, high-dimensional space. In many molecular design and drug discovery problems, Tanimoto is suitable for such high-dimensional binary problems and scale well, but it may likely not be the case in many other problem — simply fitting the GP (or other probabilistic model) on the high-dimensional space is already prohibitively expensive, not to mention any efficient optimization.

Also the optimization step described in eq (3) seems can be replaced by more established method such as Expected Improvement or Thompson sampling. If it can/should not be changed, more discussion of why should be included.

**Other Comments Or Suggestions:**

Minor: extra space in L284.

**Other Strengths And Weaknesses:**

The paper is overall well-written and easy to understand.

**Questions For Authors:**

- Instead of using the sampling strategy described in eq (3), which may intuitively make sense, but has little theoretical justification, why not use something more established such as Thompson sampling or EI to select the candidates.

- I’d love to see more discussion and/or experiment on other problem domains. As I mentioned in “Methods And Evaluation Criteria" above, the proposed method may not extend well to other domain, which may be fine, but would significantly limit its impact.

**Relation To Broader Scientific Literature:**

This is directly related to the general Bayesian Optimization and high-dimensional/latent-space BO.

**Theoretical Claims:**

N/A

---

> ### Author Rebuttal · Authors · 2025-03-31
>
> Thank you for your strongly positive review and for highlighting the novelty, clarity, and thorough experimental design of our work. We have carefully considered your two main comments, both of which guided additional discussion and improvements in the final version of the paper. These insights have helped us better refine our manuscript. We look forward to seeing how this work inspires new lines of inquiry and contributes to the continued advancement of Bayesian optimization.
>
>
> 1) **On Incorporating Arbitrary Acquisition Functions**. You raise an important question regarding the integration of an arbitrary acquisition function into COWBOYS. In essence, our proposed approach is fundamentally different from traditional BO settings that rely on standard utility-based acquisition functions. Here, we move away from maximizing an acquisition function over a constrained set of equally likely structures. Instead, we explore searching over an entire distribution (as modeled by the VAE) of likely structures.
> Because of this distribution-based view, conventional acquisition functions (e.g., Expected Improvement or Probability of Improvement) cannot be directly applied without inheriting the same limitations found in current LSBO methods, most notably their disregard for the distribution provided by the VAE. One of our paper’s main contributions is precisely to handle this distribution by replacing maximization-based strategies with a sampling-based approach (Equation 3).
> While Equation 3 does resemble Probability of Improvement (a point now greatly elaborated in the final version), adapting more sophisticated acquisition schemes is indeed an exciting direction for future research. For instance, we could condition the VAE to generate samples for which an alternative acquisition criterion (e.g., EI) exceeds a certain threshold. Determining this threshold adaptively would open up further avenues, possibly guided by theoretical regret analyses to explain COWBOYS’ strong empirical performance. We believe these avenues will become fertile ground for future studies that aim to bridge existing BO theory with generative models in complex, high-dimensional design spaces.
> 2) **Generalizability of COWBOYS Beyond Molecular Design and Drug Discovery**. Please see our response to vn27 for a discussion of the significant potential of COWBOYS in different domains.

---

### Decision · Program_Chairs · 2025-05-01

**Decision:**

Accept (spotlight poster)

**Comment:**

This paper revisits a popular approach to Bayesian optimisation in structured spaces, combining a probabilistic model for property prediction with a latent space of an autoencoder. The decoupled approach here is likely to be broadly useful in tasks where good GP surrogates exist in data space (including for many molecule and materials generation tasks). After the rebuttal period, all reviewers had a consensus that the paper should be accepted.